# PDP: A General Neural Framework for Learning SAT Solvers

## Abstract

There have been recent efforts for incorporating Graph Neural Network models for learning fully neural solvers for constraint satisfaction problems (CSP) and particularly Boolean satisfiability (SAT). Despite the unique representational power of these neural embedding models, it is not clear to what extent they actually learn a search strategy vs. statistical biases in the training data. On the other hand, by fixing the search strategy (e.g. greedy search), one would effectively deprive the neural models of learning better strategies than those given. In this paper, we propose a generic neural framework for learning SAT solvers (and in general any CSP solver) that can be described in terms of probabilistic inference and yet learn search strategies beyond greedy search. Our framework is based on the idea of propagation, decimation and prediction (and hence the name PDP) in graphical models, and can be trained directly toward solving SAT in a fully unsupervised manner via energy minimization, as shown in the paper. Our experimental results demonstrate the effectiveness of our framework for SAT solving compared to both neural and the industrial baselines.

## 1 Introduction

Constraint satisfaction problems (CSP) (Kumar, 1992) and Boolean Satisfiability (SAT), in particular, are the most fundamental NP-complete problems in Computer Science with a wide range of applications from verification to planning and scheduling. There have been huge efforts in Computer Science (Biere et al., 2009a; Knuth, 2015; Nudelman et al., 2004; Ansótegui et al., 2008; 2012; Newsham et al., 2014) as well as Physics and Information Theory (Mezard & Montanari, 2009; Krzakała et al., 2007) to both understand the theoretical aspects of SAT and develop efficient search algorithms to solve it. Furthermore, since in many real applications, the problem instances are often drawn from a narrow distribution, using Machine Learning to build data-driven solvers which can learn domain-specific search strategies is a natural choice. In that vein, Machine Learning has been used for different aspects of CSP and SAT solving, from branch prediction (Liang et al., 2016) to algorithm and hyper-parameter selection (Xu et al., 2008; Hutter et al., 2011). While most of these models rely on carefully-crafted *features*, more recent methods have incorporated techniques from Representation Learning and particularly Geometric Deep Learning (Bronstein et al., 2017; Wu et al., 2019) to capture the underlying discrete structure of the CSP (and SAT) problems.

Along the latter direction, Graph Neural Networks (Li et al., 2015; Defferrard et al., 2016) have been the cornerstone of many recent deep learning approaches to CSP – e.g., the NeuroSAT framework (Selsam et al., 2019), the Circuit-SAT framework (Amizadeh et al., 2019), and Recurrent Relational Networks for Sudoku (Palm et al., 2018). These frameworks have been quite successful in capturing the inherent structure of the problem instances and *embedding* it into traditional vector spaces that are suitable for Machine Learning models. Nevertheless, a major issue with these pure embedding frameworks is that it is not clear how the learned model works, which in turn begs the question whether the model actually learns to search for a solution or simply adapts to statistical biases in the training data. As an alternative, researchers have used deep neural networks within classical search frameworks for tackling combinatorial optimization problems – e.g. Khalil et al. (2017). In these hybrid, neuro-symbolic frameworks, deep learning is typically used to learn optimal search heuristics for a generic search algorithm – e.g. greedy search. Despite the clear search strategy, the performance of the resulted models is bounded by the effectiveness of the imposed strategy which is *not* learned.

In this paper, we take a middle way and propose a neural framework for learning SAT solvers which enjoys the benefits of both of the methodologies above. In particular, we take the formulation of solving CSPs as probabilistic inference (Montanari et al., 2007; Mezard & Montanari, 2009; Braunstein et al., 2005; Grover et al., 2018; Gableske et al., 2013) and propose a neural version of it that is capable of learning efficient inference strategy (i.e. the search strategy in this context) for specific problem domains. Our general framework is a design pattern which consists of three main operations: *Propagation*, *Decimation* and *Prediction*, and hence referred as the PDP framework. In general, these operations can be implemented either as fixed algorithms or as trainable neural networks. In this light, PDP can be seen as probabilistic inference in the latent space, and as a result, it is somewhat straightforward to establish how the search strategy in the neural PDP works, unlike pure embedding methods. On the other hand, due to the distributed nature of its decimation component, PDP is *not* restricted by the greedy strategy of the classical decimation process, meaning that it can potentially learn search strategies which are not greedy. And this would distinguishes it from the neuro-symbolic methods that are defined within the greedy strategy. Furthermore, we propose an unsupervised, fully differentiable training mechanism based on *energy minimization* which trains PDP directly toward solving SAT via end-to-end backpropagation. The unsupervised nature of our proposed training mechanism enables PDP to train on (infinite) stream of unlabeled data.

Our experimental results show the superiority of the PDP framework compared to both state-of-the-art neural and inference-based solvers. We further demonstrate our model's capability in adapting to problem distributions with distinct structure which is common in industrial settings. Lastly, we take an ambitious step further and compare our neural framework against the well-known class of Conflict-Driven Clause Learning (CDCL) industrial solvers. Despite the fact that neural solvers (including ours) are still in their infancy and not really comparable to industrial SAT solvers, we were pleasantly surprised to see that our trained models could come close to the performance of the CDCL solver. This is significant in the sense that it shows that even though our neural model at this point lacks some powerful classical search techniques such as backtracking and clause learning, its adaptive neural constraint propagation strategy is nevertheless very effective.

## 2 RELATED WORK

Classical Machine Learning has been incorporated in solving combinatorial optimization problems (Bengio et al., 2018) and SAT in particular: from SAT classification (Xu et al., 2012), and solver selection (Xu et al., 2008) to configuration tuning (Haim & Walsh, 2009; Hutter et al., 2011; Singh et al., 2009) and branching prediction (Liang et al., 2016; Grozea & Popescu, 2014; Flint & Blaschko, 2012). However, more recently, researchers have used Deep Learning to train full-stack solvers. There are two main categories of Deep Learning methodologies proposed recently. In the first category, neural networks are used to embed the input CSP instances into a latent vector space where a predictive model can be trained. Palm et al. (2018) used Recurrent Relational Networks to train Sudoku solvers. Their framework relies on provided solutions at the training time as opposed to our framework, which is completely unsupervised. Selsam et al. (2019) proposed to use Graph Neural Networks (Li et al., 2015) to embed CNF instances for the SAT classification problem. They also proposed to use a post-processing clustering approach to decode SAT solutions. In contrast, our method is fully unsupervised and is *directly* trained toward solving SAT. Amizadeh et al. (2019) proposed a DAG Neural Network to embed logical circuits for solving the Circuit-SAT problem. Their framework is also unsupervised but it is mainly for circuit inputs with DAG structure. Prates et al. (2018) proposed a convolutional embedding-based method for solving TSP.

In the second category, neural networks are used to learn useful search heuristics *within* an algorithmic search framework – typically the greedy search (Vinyals et al., 2015; Bello et al., 2016; Khalil et al., 2017), branch-and-bound search (He et al., 2014) or tree search (Li et al., 2018). While these methods enjoy a strong inductive bias in learning the optimization algorithm as well as some proof of correctness, their effectiveness is essentially bounded by sub-optimality of the imposed search strategy. Our proposed framework effectively belongs to this category but at the same time, its performance is not bounded by any search strategy.

Our framework can also be seen as learning optimal message passing strategy on probabilistic graphical models. There have been some efforts in this direction (Ross et al., 2011; Lin et al., 2015; Heess et al., 2013; Johnson et al., 2016; Yoon et al., 2018), but most are focused on merely message

passing, whereas our framework learns both optimal message passing *and* decimation strategies, concurrently.

## 3  BACKGROUND

A Constraint Satisfaction Problem, denoted by $\mathbb{CSP}\langle X, C\rangle$, aims at finding an *assignment* to a set of $N$ discrete variables $X = \{x_i : i \in 1..N\}$ each defined on a set of discrete values $\mathcal{X}$ such that it satisfies *all* $M$ constraints $C = \{c_a(\boldsymbol{x}_{\partial a}) : a \in 1..M\}$. Here, $\partial a$ is a subset of variable indices that constraint $c_a$ depends on; similarly, by $\partial i$, we denote the subset of constraint indices that variable $i$ participates in. Each constraint $c_a : \mathcal{X}^{|\partial a|} \mapsto \{0, 1\}$ is a Boolean function that takes value 1 iff $\boldsymbol{x}_{\partial a}$ satisfies the constraint $c_a$. In this paper, we focus on Boolean Satisfiability problem (SAT) where the variables take values from $\mathcal{X} = \{0, 1\}$ and each constraint (or *clause*) is a disjunction of a subset of variables or their negations.

Furthermore, any CSP instance $\mathbb{CSP}\langle X, C\rangle$ can be represented as a *factor graph* probabilistic graphical model $\mathbb{FG}\langle X, C\rangle$(Koller et al., 2009). A factor graph $\mathbb{FG}\langle X, C\rangle$ is a bipartite graph where each variable $x_i$ corresponds to a variable node in $\mathbb{FG}$ and each constraint $c_a$ corresponds to a factor node in $\mathbb{FG}$. There is an edge between the $i$-th variable node and the $a$-th factor node if $i \in \partial a$. Then, one may define a measure on FG as:

$$P(X) = \frac{1}{Z} \prod_{a=1}^{M} \phi_a(\boldsymbol{x}_{\partial a}) \tag{1}$$

where $\phi_a$ are the factor functions such that $\phi_a(\boldsymbol{x}_{\partial a}) := \max(c_a(\boldsymbol{x}_{\partial a}), \epsilon)$ for some very small, positive $\epsilon$. $Z$ is the normalization constant. In the special case of SAT, we extend the $\mathbb{FG}$ representation by assigning a binary $e_{ia} \in \{-1, 1\}$ attribute to each edge such that $e_{ia} = -1$ if variable $x_i$ appears negated in the clause $c_a$, and $e_{ia} = 1$ otherwise. This way the factor functions take the same functional form (i.e. conjunction) independent of the factor index $a$; that is, $\phi_a(\boldsymbol{x}_{\partial a}) = \phi(\boldsymbol{x}_{\partial a}, \boldsymbol{e}_{\partial a})$, where $\boldsymbol{e}_{\partial a}$ are all the edges connected to the $a$-th factor.

Using this formalism, the solutions of the original $\mathbb{CSP}\langle X, C\rangle$ correspond to the modes of $P(X)$. Given $\mathbb{FG}\langle X, C\rangle$, one can compute the marginal distribution of each variable node by probabilistic inference on the factor graph using the *Belief Propagation* (BP) algorithm (aka the Sum-Product algorithm) (Koller et al., 2009). But the actual optimization problem can be solved by computing the *max-marginals* of $P(X)$ via algorithms such as Max-Product, Min-Sum (Koller et al., 2009) and Warning-Propagation (Braunstein et al., 2005).

All of the above algorithms including BP can be seen as special cases of the General Message Passing (GMP) algorithm on factor graphs (Mezard & Montanari, 2009), where the outgoing messages from the graph nodes are computed as a deterministic function of the incoming messages in an iterative fashion. If GMP converges, at the fixed-point, the messages often contain valuable information regarding variable assignments that maximizes the marginal distributions and eventually solve the CSP. In particular, the basic procedure to solve CSPs via probabilistic inference is (1) run a specific GMP algorithm on the factor graph until convergence, (2) based on the incoming fixed-point messages to each variable node, pick the variable with the highest *certainty* regarding a satisfying assignment, (3) set the most certain variable to the corresponding value, simplify the factor graph if possible and repeat the entire process over and over until all variables are set. We refer to this process as *GMP-guided sequential decimation* or decimation for short. The most famous algorithms in this class are BP-guided decimation (Montanari et al., 2007) and SP-guided decimation, based on Survey Propagation (SP) (Aurell et al., 2005; Chavas et al., 2005).

## 4  THE PDP FRAMEWORK

In order to develop a neural framework for learning to solve SAT, first we need a suitable *design pattern* that would allow solving SAT via neural networks. To achieve this, we introduce the *Propagation-Decimation-Prediction* (PDP) framework. The PDP framework can be seen as the generalization of the GMP-guided sequential decimation procedure described in previous section, where certain restrictions are relaxed. In particular:

**(A)** In the GMP-guided sequential decimation, a decimation step is executed only after GMP is converged. We relax this requirement in PDP by interleaving propagation and decimation steps.

**(B)** In sequential decimation, at each step, only one variable is fixed (i.e. the one with the highest certainty). But as it is shown in Chavas et al. (2005), this is not necessary and multiple variables can be set concurrently in a fully distributed fashion. We follow this pattern and let the decimation step occur concurrently across the factor graph *without* any centralized selection procedure.

**(C)** In the classical decimation procedure, decimation refers to "fixing" a variable node to a certain value. In PDP, we relax this requirement in two ways: (1) we let the decimation step happen at the message-level (i.e. on the edges) rather than at the variable-level (i.e. on the nodes), and (2) instead of fixing the message on an edge to a certain value, the PDP's decimation step simply intercepts the propagator messages and *transform* them in a *stateful* manner. Note that almost all GMP-guided decimation algorithms can be expressed in terms of the PDP design pattern but not the other way around. For example, in the classical case, the decimation process is greedy by definition; whereas in PDP, that restriction has been lifted. Even though our proposed framework is versatile enough to tackle any type of CSP, in this paper, we focus on the SAT problem.

**Messages:** In the PDP framework, at each time step $t$, there are four messages defined on each edge $(i, a)$: the propagator messages in each direction, $\boldsymbol{p}_{i \to a}^{(t)}$ and $\boldsymbol{p}_{a \to i}^{(t)}$, and the decimator messages in each direction: $\boldsymbol{d}_{i \to a}^{(t)}$ and $\boldsymbol{d}_{a \to i}^{(t)}$. These messages are real vectors in a latent space $\mathbb{R}^h$.

**Propagation Step:** This step defines how the propagator messages on the edges get updated given the incoming decimator messages from the previous step. In particular for each edge $(i, a)$, we have:

$$\boldsymbol{p}_{i \to a}^{(t)} = \boldsymbol{\Psi}_\theta \big( \{ (\boldsymbol{d}_{b \to i}^{(t-1)}, e_{bi}) : b \in \partial i \setminus a \} \big), \, \boldsymbol{p}_{a \to i}^{(t)} = \boldsymbol{\Psi}_\gamma \big( \{ (\boldsymbol{d}_{j \to a}^{(t-1)}, e_{aj}) : j \in \partial a \setminus i \} \big) \quad (2)$$

where $\boldsymbol{\Psi}_\theta$ and $\boldsymbol{\Psi}_\gamma$ are general (neural network) *set functions* parametrized by the parameter vectors $\boldsymbol{\theta}$ and $\boldsymbol{\gamma}$, respectively. Similar to Amizadeh et al. (2019), in our implementation, we have used Deep Set functions (Zaheer et al., 2017) to model $\boldsymbol{\Psi}_\theta$ and $\boldsymbol{\Psi}_\gamma$. It should be emphasized that both $\boldsymbol{\Psi}_\theta$ and $\boldsymbol{\Psi}_\gamma$ are *stateless* functions that compute the outgoing propagator messages merely based on incoming decimator messages and the corresponding edge attributes.

**Decimation Step:** As mentioned before, in PDP, the decimation step simply consists of *transforming* the messages generated by the propagation step on each individual edge. Moreover, in principle, the effect of decimation usually goes beyond one iteration, which means that the decimator needs to keep a *memory* of how it transformed the same message in the previous iterations. In other words, the decimation step is inherently *stateful*. Therefore, we define the decimation step as a stateful function that calculates the decimator message on each edge based on the propagator message on the same edge as well as the decimator message at the previous iteration:

$$\boldsymbol{d}_{i \to a}^{(t)} = \boldsymbol{\Phi}_\nu (\boldsymbol{p}_{i \to a}^{(t)}, e_{ia}, \boldsymbol{d}_{i \to a}^{(t-1)}), \, \boldsymbol{d}_{a \to i}^{(t)} = \boldsymbol{\Phi}_\omega (\boldsymbol{p}_{a \to i}^{(t)}, e_{ia}, \boldsymbol{d}_{a \to i}^{(t-1)}) \quad (3)$$

Here $\boldsymbol{\Phi}_\nu$ and $\boldsymbol{\Phi}_\omega$ are recurrent neural network units (e.g. LSTM or GRU (Chung et al., 2014)) parametrized by parameter vectors $\boldsymbol{\nu}$ and $\boldsymbol{\omega}$, respectively.

**Prediction Step:** At any point during the propagation-decimation process, the model can be queried to produce the most likely (soft) assignments for the variable nodes in the factor graph. This is done via the prediction step which produces variable assignments based on the incoming decimator messages to each variable node as well as the corresponding edge attributes; that is,

$$x_i^{(t)} = \boldsymbol{\Gamma}_\zeta \big( \{ (\boldsymbol{d}_{b \to i}^{(t)}, e_{bi}) : b \in \partial i \} \big) \quad (4)$$

where $\boldsymbol{\Gamma}_\zeta$ is another deep set function neural network parametrized by the parameter vector $\boldsymbol{\zeta}$. In the SAT problem, we use the Sigmoid function as the last activation layer of $\boldsymbol{\Gamma}_\zeta$ to generate soft assignments for Boolean variables.

The tuple $\mathcal{M} = \langle \boldsymbol{\Psi}_\theta, \boldsymbol{\Psi}_\gamma, \boldsymbol{\Phi}_\nu, \boldsymbol{\Phi}_\omega, \boldsymbol{\Gamma}_\zeta \rangle$ fully specifies the PDP model. Algorithm 1 illustrates the interplay of the PDP's three steps in the forward path. Note that at the train time, the forward computation returns back one set of assignments per each iteration step – i.e. a total of $T_{max}$ sets of soft assignments. This is mainly done for the loss computation purposes, as shown in the next section. At the test time, however, the iteration loop terminates as soon as a satisfying assignment is found (Line 13 in Algorithm 1). In order to check whether a soft assignment is SAT, we first apply the thresholding function $\delta(x_i^{(t)}) = I(x_i^{(t)} > 0.5)$ on soft assignments to get the hard Boolean assignment for each variable.

---

**Algorithm 1:** The PDP Forward Computation

---

**input :** Factor graph $\mathbb{FG}\langle X, C \rangle$ and $T_{max}$

1   Randomly initialize $\boldsymbol{p}_{i \to a}^{(0)}, \boldsymbol{p}_{a \to i}^{(0)}, \boldsymbol{d}_{i \to a}^{(0)}, \boldsymbol{d}_{a \to i}^{(0)}$

2   **for** $t = 1$ **to** $T_{max}$ **do**

3     `// Propagation step`

4     **foreach** $(i, a) \in \mathbb{FG}\langle X, C \rangle$ **do**

5       Compute $\boldsymbol{p}_{i \to a}^{(t)}, \boldsymbol{p}_{a \to i}^{(t)}$ using Eq. (2)

6     `// Decimation step`

7     **foreach** $(i, a) \in \mathbb{FG}\langle X, C \rangle$ **do**

8       Compute $\boldsymbol{d}_{i \to a}^{(t)}, \boldsymbol{d}_{a \to i}^{(t)}$ using Eq. (3)

9     `// Prediction step`

10    **for** $i = 1$ **to** $N$ **do**

11      Compute $x_i$ using Eq. (4)

12    $X^{(t)} \leftarrow \{x_i : i \in 1..N\}$

13    **if** *Testing* **and** $\delta(X^{(t)})$ *is SAT* **then**

14      **return** $X^{(t)}$

15   **return** $\{X^{(t)} : t \in 1..T_{max}\}$

---

Finally note that, at first sight, the PDP framework bears a resemblance to the NeuroSAT model in Selsam et al. (2019) in the sense that both methods produce latent representations on a bipartite graph via some notion of "message passing". Nevertheless, we would like to emphasize that these methods are fundamentally different. In NeuroSAT, the latent representations are defined per each node and represent the node embeddings of the bipartite representation of CNF, whereas in PDP, the latent vectors represent messages per each *directed* edge. Also, the message passing in NeuroSAT refers to the process proposed in Li et al. (2015) for graph embedding, whereas in our framework, message passing refers to belief propagation on the factor graph graphical model.

**Parallelization and Batch Replication:** Our implementation of PDP in PyTorch is embarrassingly parallel: every inner loop in Algorithm 1 runs concurrently on all the edges/nodes of the input factor graph. Furthermore, parallel processing of multiple input factor graphs in a single batch is straightforward in PDP: we simply *concatenate* all the instances in a batch into one large factor graph and treat the result as one CNF formula with multiple independent clauses. Using this powerful trick allows us to solve many examples simultaneously on GPU.

Furthermore, batch parallelization can be also used to expedite the search in single problem scenarios. Note that a CSP may have many solutions and which one of them is found by PDP depends on the random initialization in Line 1 of Algorithm 1. Therefore, at the test time, we can *replicate* each single example in the batch multiple times knowing that each replica will be initialized by different random values; then, the iteration loop terminates as soon as the solver finds a solution for at least one of the replicas. This process, referred as *batch replication*, enables the solver to simultaneously search different parts of the solution space for a single problem.

## 5   Training a PDP SAT Solver

In order to train a PDP model $\mathcal{M}$ to solve SAT problems, we would ideally like to reward the model outputs that have high probability regarding the measure in Eq. (1). As a result, one can train $\mathcal{M}$ by maximizing the probability of the model output w.r.t. the model parameters. Instead, a more numerically stable method aims at minimizing the negative log-likelihood aka the *energy function*:

$$\mathcal{E}(X) = -\log P(X) = \log Z - \sum_{a=1}^{M} \log \phi(\boldsymbol{x}_{\partial a}, \boldsymbol{e}_{\partial a}) \tag{5}$$

Nevertheless, $\phi(\cdot)$ is not a differentiable function as we defined it in Section 3. Moreover, $\mathcal{M}$ produces soft assignments which cannot be directly fed into $\phi(\cdot)$. To resolve these issues, for training purposes only, we define a differentiable proxy to the original $\phi(\cdot)$. In particular, in the SAT problem, $\phi(\cdot)$ should encode the logical disjunction on soft assignments. One possible differentiable formulation is the *smooth max* function as proposed by Amizadeh et al. (2019):

$$S_{max}(\boldsymbol{x}_{\partial a}, \boldsymbol{e}_{\partial a}) = \frac{\sum_{i \in \partial a} e^{\ell(x_i, e_{ia})/\tau} \ell(x_i, e_{ia})}{\sum_{i \in \partial a} e^{\ell(x_i, e_{ia})/\tau}}, \text{ where } \ell(x_i, e_{ia}) = \begin{cases} x_i & \text{if } e_{ia} = 1 \\ 1 - x_i & \text{otherwise} \end{cases} \tag{6}$$

is the *literal function*, and $\tau$ is the temperature parameter. Similar to Amizadeh et al. (2019), we start the training at high a temperature and gradually anneal it toward 0. By doing so, we effectively let $S_{max}(\cdot)$ start off as the arithmetic mean function and gradually turn into the $\max(\cdot)$ function, which is the equivalent of logical disjunction on soft assignments. In order to mimic the behavior of disjunction even further, we use the smooth Step function to enhance the *contrast* of the $S_{max}(\cdot)$

output by pushing the soft assignment values to the extremes depending on whether they are below 0.5 or above it:

$$G_\kappa(x) = \frac{x^\kappa}{x^\kappa + (1-x)^\kappa} \qquad (7)$$

where $\kappa > 1$ is a constant. Using Eqs. (6), (7), finally we can define our smooth proxy for $\phi(\cdot)$ as $\phi(\boldsymbol{x}_{\partial a}, \boldsymbol{e}_{\partial a}) = G_\kappa\big(S_{max}(\boldsymbol{x}_{\partial a}, \boldsymbol{e}_{\partial a})\big)$. Given this proxy, we define the loss function for the example $\mathbb{FG}\langle X, C \rangle$ as the discounted accumulated energy over $T_{max}$ iterations:

$$\mathcal{L}_\lambda\big(\mathbb{FG}\langle X, C \rangle\big) = \sum_{t=1}^{T_{max}} \lambda^{(T_{max}-t)} \mathcal{E}(X^{(t)}) \qquad (8)$$

where $\{X^{(t)} : t \in 1..T_{max}\}$ is the output of model $\mathcal{M}$ according to Algorithm 1, and $0 < \lambda \le 1$ is the *discounting factor*. The idea here is the model is penalized more if it produces non-SAT assignments further down the iteration loop. Finally, we note that using this loss function, our training methodology is completely unsupervised; that is, it does *not* need SAT solutions at the training time nor does it need SAT/UNSAT binary labels. And yet, the proposed mechanism *directly* trains the PDP model toward solving SAT. In that respect, our framework is very different from the NeuroSAT framework (Selsam et al., 2019) where solver is built indirectly via training a binary SAT classifier.

## 6   EXPERIMENTAL RESULTS

We have compared our PDP framework against three different categories of baselines: (a) the classical probabilistic inference based techniques for SAT solving, (b) the state-of-the-art neural embedding SAT solver and (c) a CDCL industrial SAT solver, as follows[1]:

**SP**: Survey-propagation guided decimation (Mezard & Montanari, 2009) is one of the most effective inference-based algorithms to approach random uniform $k$-SAT problems in the *hard SAT region*.

**Reinforce**: Unlike PDP, the decimation process in SP is sequential. Not only can this slow down the search but also it restricts the search to the greedy strategy. To address these issues, Chavas et al. (2005) have proposed a fully distributed version of SP called the *Reinforce Algorithm*.

**NeuroSAT**: NeuroSAT (Selsam et al., 2019) is the state-of-the-art neural method based on learning a SAT classifier.

**Glucose**: Glucose (Audemard & Simon, 2018) is a popular Conflict-Driven Clause Learning (CDCL) SAT solver for industrial problems (Biere et al., 2009b) whose basic techniques have become common practice for many modern SAT solvers.

Of these baselines, SP and Reinforce are implemented using the same PDP platform. All the PDP-based methods as well as NeuroSAT take a maximum iteration number (i.e. $T_{max}$) which controls the timeout budget; in our experiments, we have set $T_{max} = 1000$. For Glucose, we have incorporated explicit timeout policy since it does not support controllable iteration number.

In our experiments, the dimension of the message space ($h$) is set to 150. The deep set functions $\Psi_\theta$, $\Psi_\gamma$, and $\Gamma_\zeta$ are implemented according to the formulation in Theorem 2 in Zaheer et al. (2017), where $\rho$ and $\phi$ are each 2-layer Perceptrons. The decimator functions $\Phi_\theta$ and $\Phi_\gamma$ are implemented as GRU cells. The LogSigmoid function is used for all internal non-linearities. We have used the Adam optimizer with learning rate of $10^{-4}$ and gradient clipping with norm 0.65 to train our model. We have also enforced weight decay of $10^{-10}$ as well as dropout with rate 0.2 for regularization. These hyper-parameters are picked using grid-search on a training dataset with smaller variable sizes (up to 20) over 5 epochs. For NeuroSAT, we have used the default settings proposed in Selsam et al. (2019).

### 6.1   UNIFORM RANDOM $k$-SAT

Uniform random $k$-SAT problems (where each clause in the input CNF has exactly $k$ literals and is selected *uniformly* from a set of variables) have been studied in depth in Combinatorial Optimization,

---

[1]We have provided our PyTorch implementation accompanied by detailed documentation and the config files for the neural PDP as well as the PDP-based SP and Reinforce baselines in the supplemental materials.

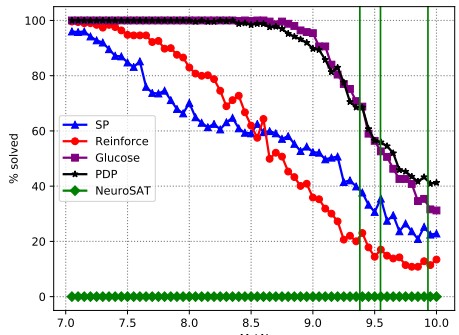 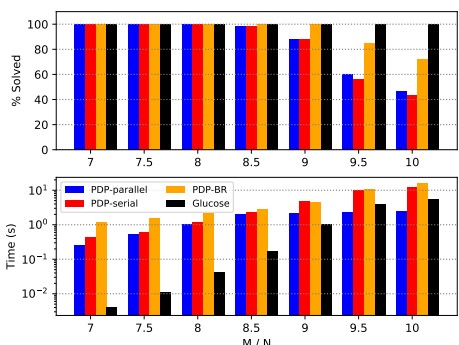

Figure 1: **(Left)** The ratio of the uniform 4-SAT problems solved versus $\alpha$. The green vertical lines indicate the theoretical phase transition thresholds for uniform 4-SAT as $N \to \infty$. **(Right)** The accuracy and time for Glucose vs. PDP (parallel), PDP (serial) and PDP (serial with batch replication) on uniform 4-SAT. All PDP methods were run up to $1000$ iterations, whereas Glucose was let to solve all the problems without time restriction.

Statistical Physics and Coding Theory (Mezard & Montanari, 2009). In particular, researchers have rigorously identified four different phases of complexity most random $k$-SAT problems go through as the ratio of clauses to variables (known as $\alpha \equiv M/N$) grows: the replica-symmetric (RS) phase (aka the easy SAT phase), the dynamical phase, the condensation phase and finally the UNSAT phase (Krzakała et al., 2007).

**Training**: Since the training strategy in Section 5 is fully unsupervised, at the training time, we simply generate a stream of unlabeled examples in memory to train a PDP SAT solver. For the training examples, we let the size of each clause in a CNF varies randomly between 2 and 10. Also the $\alpha$ value for each CNF varies between 2 and 10, while the number of variables $N$ is chosen uniformly between 4 and 100. This guarantees that our training stream will contain all ranges of problems in terms of complexity. Training of NeuroSAT, on the other hand, is based on training of a supervised SAT/UNSAT classifier which not only does need binary labels, but also imposes a strict training regime to avoid learning superfluous features, as described in Selsam et al. (2019). As a result, we have adopted the same regime to generate offline training data for NeuroSAT. Nevertheless, we were not able to train a NeuroSAT classifier for problems with more than 20 variables, even when we increased the model capacity. Therefore, the NeuroSAT model is only trained with instances of 4 to 20 variables instead.

**Results**: We have generated test datasets of uniformly random satisfiable 4-SAT problems (each with 100 variables) with various $\alpha$ values according to the process described in Mezard & Montanari (2009) chapter 10. In particular, for any given value of $\alpha$, a dataset of $500$ SAT examples has been generated where all examples in the dataset have roughly the same $\alpha$ value. This process was then repeated for different values of $\alpha$ to cover all the aforementioned phases. Figure 1(Left) shows the results of running the models on the 4-SAT test datasets. All the PDP-based models as well as NeuroSAT are run for $T_{max} = 1000$ iterations. This would translate to 3s per example timeout threshold for Glucose. As the plot shows, NeuroSAT model could not solve any of the problems even though its underlying SAT/UNSAT classifier reached a small training error. We suspect this might be due to the fact that we were not able to train NeuroSAT on larger problems and the fact that it is not directly trained to solve SAT to begin with. Our neural PDP method, on the other hand, significantly outperformed the SP and Reinforce algorithms signifying the importance of learning compared to classical (fixed) inference methods. Moreover, within the 3s timeout budget for Glucose, our framework performs as par with Glucose. This may not seem fair because unlike Glucose, PDP processes multiple examples in a batch each time. To further investigate this, we have let Glucose to go beyond 3s timeout limit and compared it against the original parallel PDP, its serial version and its serial version augmented by batch replication (each run for $1000$ iterations). Figure 1(Right) shows the accuracy and time comparison results. From this plot, we can see that: (1) the serial versions of PDP cannot beat Glucose, (2) Glucose can eventually solve all the problems in the datasets if given opportunity to go beyond 3s budget, and (3) batch replication significantly improves the accuracy of neural PDP while time overhead is roughly similar to that of serial PDP. The latter observation is

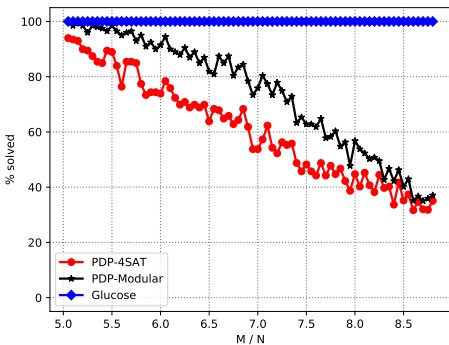

Figure 2: The ratio of the modular 4-SAT problems solved versus $\alpha$ for Glucose and the two PDP solvers trained on uniform and modular random 4-SAT.

crucial because it shows combined with batch replication, PDP can achieve even higher accuracy within limited time budget. This further shows the potential of combining a pure neural framework with classical *restart* mechanism, which is introduced to PDP via batch replication.

### 6.2 PSEUDO-INDUSTRIAL RANDOM $k$-SAT

As mentioned earlier, one of the main goals of incorporating Machine Learning in combinatorial optimization is to arrive at generic, data-driven solvers that can adapt to different problem domains. Nevertheless, the neural SAT solvers proposed in the literature so far are mainly trained on uniform random $k$-SAT problems. This begs the question whether neural solvers are capable of picking up domain-specific information inherent in a narrow problem distribution. To this end, in this section, we go beyond uniform $k$-SAT and focus on problems that exhibit distinct structures. In particular, we consider pseudo-industrial random SAT problems with highly *modular* structures (Newman, 2006). This is mainly motivated by the fact that real industrial SAT problems are highly modular (Ansótegui et al., 2012). To generate such problems, we have incorporated the *Community Attachment* (CA) model proposed in Giráldez-Cru & Levy (2016).

**Training**: We have trained two neural PDP models with the same exact architecture and capacity as the one in the previous section. The first model has been trained on a stream of uniformly generated 4-SAT problems with the number of variables ranging from 5 to 100. The second model, however, is trained on a stream of pseudo-industrial random 4-SAT problems generated according to the CA model. Each random example in the CA stream has between 10 to 20 communities with modularity factor $Q$ between 0.8 and 0.9. We note that the $Q$ value does not typically go beyond 0.3 for uniform $k$-SAT problems (Ansótegui et al., 2012).

**Results**: We have tested both the trained PDP models as well as Glucose on the datasets of modular 4-SAT problems generated with the same setting as the modular training data for different values of $\alpha$ – i.e. the CA model in Giráldez-Cru & Levy (2016). Figure 2 shows the performance of these models as $\alpha$ grows. Each point in the plot represents a dataset of 200 modular examples with the corresponding $\alpha$ value. The PDP models are run for $T_{max} = 1000$ iterations which translates to 2s per example timeout budget for Glucose. As the plot shows, Glucose beats the PDP solvers by solving all the problems for all $\alpha$ values within the time budget. This is somewhat expected as CDCL solvers have been shown to exploit the modular structure (Newsham et al., 2014; Giráldez-Cru & Levy, 2016) and as a result perform very well on real-world industrial problems. However, among the neural PDP models, the one that has been trained on modular 4-SAT problems perform significantly better than the one trained on uniform 4-SAT problems. This is an important result because classical message passing inference algorithms (like SP) work reasonably well if the underlying graph is *locally tree-like* (Mezard & Montanari, 2009), which holds for uniform random $k$-SAT problems as $N \to \infty$, but does not generally hold for non-uniform random graphs. Nevertheless, the neural PDP can still learn efficient, domain-specific inference strategies. This indicates that the neural PDP framework has the potential to adapt to domains beyond uniform random $k$-SAT and be used as a generic yet adaptive solver.

We also note that the gap between the two models shrinks for high $\alpha$ values. This phenomenon can be explained as follows: intuitively, in modular formulas, solving each module (which is roughly a uniform $k$-SAT) is a necessary (not sufficient) condition for solving the entire formula; this means that the hardness of individual modules puts a lower bound on the hardness of the formula. This is not problematic for medium-range $\alpha$ and so the gap between the PDP trained on modular problems and the uniform PDP is significant ($\sim 20\%$). But, for high $\alpha$'s, since the lower bound is already pretty high, it would effectively make a modular problem as hard as a uniform problem of same $\alpha$ for message-passing decimation algorithms and as result the gap between the two models shrinks.

## 7 CONCLUSIONS AND FUTURE DIRECTIONS

In this paper, we proposed the neural PDP framework for learning neural SAT solvers. Unlike recent methods in the literature which are based on learning efficient embeddings, our framework can be interpreted as a non-trivial neural extension of probabilistic message passing and inference techniques on graphical models and as such its search strategy can be viewed in the probabilistic terms. Furthermore, we proposed a fully unsupervised training procedure based on the idea of energy minimization on graphical models to train PDP toward solving SAT. Due to its unsupervised nature, this training mechanism enables us to train PDP on an (infinite) stream of unlabeled problem instances generated in real-time. This is in stark contrast with strict supervised training mechanism taken by some recent frameworks such as NeuroSAT. This also opens the door to the further question of how to generate unlabeled training streams for efficient training in specific domains.

We also note that PDP as a general design pattern is a powerful idea on its own as it allows many classical message passing algorithms such as SP, BP and Reinforce to be reformulated in terms of a fully parallel framework which can be run on GPUs. Moreover, the general PDP framework does *not* require its components to be necessarily neural. This gives rise to *hybrid* neuro-symbolic solvers where some components of the model are learned while others are fixed (e.g. using a SP propagator with a neural decimator). We leave the exploration this aspect of PDP to the future work.

Since neural PDP actually learns an inference algorithm, it can find multiple solutions for an input problem depending on its messages' initial values. This further gives rise to the idea of batch replication which in turn introduces the classical notion of restarts in the PDP framework. In this direction, an important future work is to investigate ways of introducing other classical search techniques used in CSP solvers, specifically backtracking and clause learning.

Our experimental results on $k$-SAT problems showed the effectiveness of the PDP framework compared to neural and industrial baselines. In particular, we saw that not only did PDP outperform classical baselines but it also came close to the well-known Glucose solver. Moreover, we showed that PDP can go beyond uniform problems and adapt to pseudo-industrial problem domains with prominent modular structures. Finally, we would like to emphasize that neural PDP is still far from claiming victory over the industrial solvers. However, the fact that our PyTorch prototype could outperform other classical and neural solvers and come close to an industrial solver reveals the huge potential in pursuing this direction, especially regarding translating other classical search techniques used in CDCL and other industrial solvers into neural frameworks such as PDP.

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
