# OpenReview forum: "PDP: A General Neural Framework for Learning SAT Solvers"
_ICLR.cc/2020/Conference — Reject_

### Official Review · AnonReviewer3 · 2019-10-21
**Official Blind Review #3**

**Rating:** 1

**Review:**

This paper investigates the well-studied problem of solving satisfiability problems using deep learning approaches. In this setting, the authors propose a neural architecture inspired by message passing operations in deep probabilistic graphical models. Namely, the architecture takes as input a CNF formula represented as a factor graph, and returns as output a set of soft assignments for the variable nodes in the graph. The internal layers of the architecture consist of propagation, decimation and prediction steps. Notably, decimation operations take an important role in learning non-greedy search strategies. Besides PDP operations, the architecture incorporates parallelization and batch replication techniques. The learning model is trained in a non-supervised way, using a cumulative (discounted) log-likelihood loss that penalizes the non-satisfying assignments returned by the algorithm.

Overall, the paper is relatively well-written and well-positioned with respect to related work. However, it is difficult to accept the paper in its current state: as explained below, it is difficult to understand how the PDP architecture effectively solves SAT problems, and experiments are not really conclusive.

Although the idea of using general message-passing techniques for learning to solve SAT problems is relevant, the overall architecture left me somewhat confused. In the SAT problem, we have a CNF formula, say $F$, and the goal is to predict whether $F$ is satisfiable or not. In the former case, the solver is required to additionally supply a model of $F$, that is, an assignment of variables to values satisfying $F$. However, unless I missed something, the PDP architecture returns as output a set of $T$ “soft assignments” for each input SAT instance, which leads to two major concerns:
* There is no final decision (SAT/UNSAT), so how can we predict the satisfiability of an instance $F$ with just a set of $T$ assignments? I guess that the PDP model will predict SAT (resp. UNSAT) if the resulting loss is small (resp. large) enough, but this is very unclear.
* Furthermore, the output set consists of “soft” assignments, as defined by (4). But a solver should return a "discrete" assignment mapping variables to values in $\{0,1\}$. So, how can we convert soft assignments to discrete ones? Are the authors using a rounding method?

The experimental results are a bit confusing too. First of all, which generator has been used for random instances ($4$-SAT) and pseudo-industrial instances? For the sake of reproducibility, this should be mentioned in the revised version of the paper. Furthermore, it seems that at first sight, PDP is competitive with Glucose, as illustrated in the left part of Figure 1. Notably, the performance of Glucose degrades as the ratio $M/N$ grows. But this is not surprising because its timeout is set to 3s. However, the right part of Figure 1 is telling another story: apparently, Glucose can solve all instances in less than 10s. So, for the sake of fairness, it would be legitimate to report curves (left part) using 10s per instance: this would highlight the behavior of PDP with respect to modern SAT solvers on random instances using reasonable timeouts for the UNSAT part. The idea of making experiments on pseudo-industrial instances is interesting, but the PDP algorithm trained on those instances (i.e. PDP-modular) is rapidly degrading as the ratio $M/N$ increases. In fact, the difference between PDP-4SAT and PDP-modular is not statistically significant for UNSAT instances. Finally, Glucose is clearly dominating PDP on pseudo-industrial instances, as it can solve all instances with just a timeout of 2s per instance.

**Experience Assessment:**

I have read many papers in this area.

**Review Assessment: Checking Correctness Of Derivations And Theory:**

I assessed the sensibility of the derivations and theory.

**Review Assessment: Checking Correctness Of Experiments:**

I assessed the sensibility of the experiments.

**Review Assessment: Thoroughness In Paper Reading:**

I read the paper at least twice and used my best judgement in assessing the paper.

---

> ### Author Response · Authors · 2019-11-08
> **Response to Reviewer #3**
>
> We appreciate the reviewer's detailed and in-depth comments. Nevertheless, it seems there are some major misunderstandings regarding how the PDP works as well as the experimental setup and the claims made in the paper. In the following, we are hoping to clarify these confusions about our work.
>
> A) As for how PDP predicts and produces variable assignments for a given CNF problem, we'd like to emphasize that PDP outputs variable assignments slightly differently depending on whether it's in the training mode or the testing mode:
>
> - In the training mode, PDP produces a set of T soft assignments which are respectively used to evaluate the loss function in Eq. (8). Although the energy function in Eq. (8) is a differentiable proxy for satisfiability, we do NOT need to explicitly check whether (the hard version of) the soft assignment actually satisfies the input formula. All we care about during training is to evaluate the loss function in Eq. (8).
>
> - In the testing mode, the algorithm returns ONLY ONE soft assignment (and not a set) -- i.e. the first soft assignment it finds whose hard version satisfies the input formula. This is done via the If-statement in Lines 13-14 in Algorithm 1. Also, in order to find the hard version of a soft assignment, we simply threshold it at 0.5.
>
> We'd like to emphasize that we NEVER try to predict SAT vs. UNSAT anywhere in the PDP architecture; that is, the Prediction step in PDP is trained to directly output variable assignments and NOT the SAT/UNSAT classification. In fact, this is one of the major differentiating factors of our work from the previous work in the NeuroSAT framework where the model is trained to predict SAT vs UNSAT. Our model, on the other hand, is directly trained to find and output satisfying variable assignments. We have made this more clear in the revised version of the paper.
>
> B) As for the generators used in the experiments, we have made it more clear in the revised version, but here they are again:
>
> - For uniform random k-SAT, we have used the generation process described in Mezard & Montanari (2009) chapter 10 for generating the evaluation data. For training data, we have described our generation process in Section 6.1 in details.
>
> - For training NeuroSAT, we have used the generation process proposed in Selsam et al (2019).
>
> - For generating modular training/evaluation data, we have used the Community Attachment (CA) process proposed in Giráldez-Cru & Levy (2016).
>
> C) As for comparing against Glucose, we have clearly stated in the paper that we make no claims regarding outperforming Glucose, on the contrary, we are far from it. This is mainly because pure neural SAT solvers like NeuroSAT and ours still lack some fundamental operations like backtracking which enable CDCL solvers to dramatically reduce the search space for a SAT solution. Nevertheless, we've found it crucial to examine where our model (and neural models in general) is standing compared to industrial solvers like Glucose.
>
> As for the timeout threshold, note that the 3s timeout is not arbitrary but rather is chosen to keep the overall execution time for Glucose roughly equal to that of the other models, as stated in the paper. But then again, as the reviewer mentioned, this can be unfair to Glucose since it can eventually solve all the problems. And because of this, we have run a second experiment without any timeout and included the results in Fig. 1 (Right). The second experiment not only shows the superiority of Glucose in the untimed settings but also it shows how batch replication can push PDP closer to Glucose's performance.
>
> D) As for converging behaviors of PDP-4SAT and PDP-modular in the UNSAT region, we have given an explanation for this phenomenon in the last paragraph of Section 6.2. In a nutshell, the hardness of a modular problem is lower-bounded by the hardness of each (dense) module individually. This means that when M/N significantly increases into the UNSAT region, solving the entire problem is lower-bounded by solving over-constrained modules which are already pretty hard to solve for pure message-passing algorithms (such as ours, NeuroSAT, SP, Reinforce, etc.) This means that in the UNSAT region, there's not that much difference between solving an over-constrained module vs an over-constrained uniform 4SAT using pure message-passing, because both have the same high density M/N. However, in the medium-range M/N, since the hardness lower-bound is not too tight, PDP-Modular has some wiggle room to learn domain-specific search strategies which subsequently leads to 20% boost over PDP-4SAT and this is statistically significant.

---

### Official Review · AnonReviewer2 · 2019-10-23
**Official Blind Review #2**

**Rating:** 6

**Review:**

The authors develop an unsupervised method for solving SAT problems. The method consists of an energy-based loss function which is optimized by a three-stage architecture that performs propagation, decimation, and prediction (PDP). The authors show that on uniform random 4-SAT problems, their PDP system outperforms two classical methods, a prior neural method, and performs favorably in comparison to a heavily developed industrial solver. Further, they show that a PDP system trained on modular 4-SAT problems performs better on modular 4-SAT problems that one trained on uniform random 4-SAT problems.

This well-written paper introduces an appealing unsupervised method for learning solvers for an important class of problems. Their results seem to be much stronger than the prior neural state-of-the-art, which also has the downside of requiring labelled data. The authors argue convincingly that although their method does not outperform Glucose, it still constitutes an useful advance in the use of neural methods for solving CSPs.

Overall, this seems to me like a useful contribution. Currently my accept recommendation is weak only because I'm not familiar enough with this area to verify that there are not other prior-work comparisons that should have been included.

Comments
The modular 4-SAT experiment is a bit underwhelming. This is not necessarily the fault of the learning algorithm: it may be that modular k-SAT problems are not the best setting to showcase the potential benefits of learning domain-specific solvers.

I don't often see field names such as "Machine Learning" and "Deep Learning" capitalized as they are in this paper.

Typos
In contract -> In contrast
N discrete variable X -> N discrete variables X?

**Experience Assessment:**

I do not know much about this area.

**Review Assessment: Checking Correctness Of Derivations And Theory:**

N/A

**Review Assessment: Checking Correctness Of Experiments:**

I assessed the sensibility of the experiments.

**Review Assessment: Thoroughness In Paper Reading:**

I made a quick assessment of this paper.

---

> ### Author Response · Authors · 2019-11-08
> **Response to Reviewer #2**
>
> We thank the reviewer for their positive feedback and the points raised. We agree with the reviewer that choosing modular k-SAT problems to demonstrate the domain-adaptability of the PDP framework is somewhat challenging mainly due to the fact that pure message-passing algorithms (without backtracking) are not known to be the best choice for modular problems. We could have, in theory, demonstrated the domain-adaptability of PDP by slightly changing the parameters of the random generation process for uniform k-SATs as the new domain. But that would be much less realistic scenario compared to modular problems which are quite common in real industrial problems. Therefore, in the interest of practicality, we have chosen the much more realistic modular problems in our experiments. Nevertheless, we would like to highlight that even with more challenging modular problems, the domain-adapted PDP shows up to 20% improvement compared to the vanilla PDP for medium range problems, which is quite significant.

---

### Official Review · AnonReviewer1 · 2019-10-26
**Official Blind Review #1**

**Rating:** 3

**Review:**

The paper presents an approach, PDP, to solve Boolean satisfiability (SAT) by decomposing it into Propagation, Decimation and Prediction, where each can be learned with a neural network.

Strength:
-	The proposed approach makes sense to me and allows modularity.
-	The paper compares the approach to several prior works including the recent NeuroSAT and Glucose, a Conflict-Driven Clause Learning (CDCL) SAT solver for industrial problems. Surprisingly, NeuroSAT cannot handle the problems studied in this work. The proposed PDP performs similar to Glucose on the studied problems.
-	The paper is clearly written and seem over all solid.

Weaknesses:
1.	Comparison to prior work
1.1.	To allow a better comparison to prior work, I am wondering why the author did not compare in a setting and dataset prior work evaluated SAT solvers.
1.2.	It would be interest to know if and how the proposed model performs on the problems evaluated in Selsam 2019.
2.	The paper could be improved by including better ablations to understand where the strength comes from, specifically w.r.t. to the design choices of Propagation, Decimation and Prediction and the relation to Selsam 2019.

An overall solid paper, which could be improved by better comparison to prior work, by using setups used previously and better analyzed by providing clear ablations which allow better understanding the individual components of the system.
I am borderline on this paper, also given my limited knowledge of the field.



**Experience Assessment:**

I do not know much about this area.

**Review Assessment: Checking Correctness Of Derivations And Theory:**

I did not assess the derivations or theory.

**Review Assessment: Checking Correctness Of Experiments:**

I assessed the sensibility of the experiments.

**Review Assessment: Thoroughness In Paper Reading:**

I read the paper at least twice and used my best judgement in assessing the paper.

---

> ### Author Response · Authors · 2019-11-08
> **Response to Reviewer #1**
>
> We appreciate the thoughtful review; below, we will address some of the concerns raised by the reviewer:
>
> A) To the best of our knowledge, Selsam et al (2019) have not released any of their training/evaluation synthetic datasets they used in their paper. Instead, they have published the exact random process for generating the data as well as the code for it. We have used their code to generate training data to train the NeuroSAT model in our experiments. As for evaluation data, the random problem instances used for evaluation in the NeuroSAT paper only covers a narrow range of \alpha = M/N. Whereas in our experiments, we wanted to evaluate the competing models on the full range of \alpha's (from the easy replica-symmetric (RS) phase all the way to the super difficult UNSAT phase, as described in the paper). In other words, the evaluation range used in Selsam et al (2019) for evaluation is already included in our range of evaluation and only constitutes a narrow part of it.
>
> B) As for the ablation study of different steps of the PDP architecture, we agree with the reviewer that it'd be insightful to know how much each step contributes to the overall performance of the model. Nevertheless, we'd like to emphasize that even though the clear separation of the steps in the PDP design pattern is more interpretable and therefore more preferable compared to the monolithic architecture of NeuroSAT, we strongly believe that the true advantage of the PDP architecture comes from its loss function and its training scheme. That is, PDP is directly trained toward solving SAT problems whereas NeuroSAT is trained toward SAT binary classification and finding a SAT solution is only a secondary process which is *not* directly targeted by its learning algorithm. In other words, NeuroSAT is not exactly optimized to find SAT solutions but rather to classify input problems as SAT vs. UNSAT, whereas our model directly targets the former.

---

### Decision · Program_Chairs · 2019-12-19

**Decision:**

Reject

**Comment:**

The authors present a neural framework for learning SAT solvers that takes the form of probabilistic inference. The whole process consists of propagation, decimation and prediction steps, so unlike other prior work like Neurosat that learns to predict sat/unsat and only through this binary signal,  this work presents a more modular approach, which is learned via energy minimization and it aims at predicting assignments (the assignments are soft which give rise to a differentiable loss). On the other hand, at test time the method returns the first soft assignment whose hard version (obtained by thresholding) satisfies the formula.  Reviewers found this to be an interesting submission, however there were some concerns regarding (among others) comparison to previous work.

Overall, this submission has generated a lot of discussion among the reviewers (also regarding how this model actually operates)  and it is currently borderline without a strong champion. Due to the concerns raised and the limited space in the conference's program, unfortunately I cannot recommend this work for acceptance.